# The Relationship of Tree Nuts and Peanuts with Adiposity Parameters: A Systematic Review and Network Meta-Analysis

**DOI:** 10.3390/nu13072251

**Published:** 2021-06-30

**Authors:** Rubén Fernández-Rodríguez, Arthur E. Mesas, Miriam Garrido-Miguel, Isabel A. Martínez-Ortega, Estela Jiménez-López, Vicente Martínez-Vizcaíno

**Affiliations:** 1Health and Social Research Center, Universidad de Castilla La-Mancha, 16071 Cuenca, Spain; Ruben.Fernandez@uclm.es (R.F.-R.); Miriam.Garrido@uclm.es (M.G.-M.); IsabelA.Martinez@uclm.es (I.A.M.-O.); Estela.JimenezLopez@uclm.es (E.J.-L.); Vicente.Martinez@uclm.es (V.M.-V.); 2Health Science Centre, Universidade Estadual de Londrina, Londrina 86038-350, Brazil; 3Facultad de Enfermería, Universidad de Castilla-La Mancha, 02006 Albacete, Spain; 4Facultad de Ciencias de la Salud, Universidad Autónoma de Chile, Talca 1101, Chile

**Keywords:** obesity, dietary, body composition, overweight

## Abstract

The network meta-analysis and systematic review conducted aim to comparatively assess the effects of tree nuts and peanuts on body weight (BW), body mass index (BMI), waist circumference (WC), and body fat percentage (BF%). A systematic search up to 31 December 2020 was performed. A random-effects network meta-analysis was conducted following the PRISMA-NMA statement. A total of 105 randomized controlled trials (RCTs) with measures of BW (n = 6768 participants), BMI (n = 2918), WC (n = 5045), and BF% (n = 1226) were included. The transitivity assumption was met based on baseline characteristics. In the comparisons of nut consumption versus a control diet, there was no significant increase observed in any of the adiposity-related measures examined except for hazelnut-enriched diets, which raised WC. Moreover, almond-enriched diets significantly reduced WC compared to the control diet and to the pistachio-, mixed nuts-, and hazelnut-enriched diets. In subgroup analyses with only RCTs, designed to assess whether nut consumption affected weight loss, almonds were associated with reduced BMI and walnuts with reduced %BF. The evidence supports that: (1) tree nut and peanut consumption do not influence adiposity, and (2) compared to a control diet, the consumption of almond-enriched diets was associated with a reduced waist circumference.

## 1. Introduction

The increase in the prevalence of overweight and obesity that has been observed since the 1980s has made excess weight a major public health concern [1]. Obesity is a multifaceted chronic disease characterized by an excess of body fat that impairs health and is associated with an increased risk of complications that reduce life expectancy, including hypertension, type 2 diabetes, cardiovascular diseases (CVDs), dyslipidemia, and cancer [2]. Some multidimensional approaches have been proposed for the prevention and management of overweight and obesity. Most are based on increasing physical activity and improving diet quality [3,4,5] to counteract energy imbalance and, thereby, avoid weight gain [6]. In this context, in line with dietary guidelines aimed at reducing total energy intake and improving diet quality, recent studies have reinforced the importance of replacing the intake of unhealthy foods between main meals for nutrient-dense foods, such as nuts [7,8,9].

According to its botanical definition, a nut is a dried fruit in which the ovary walls are very hard at maturity and the seed is unattached or free within the ovary wall [10]. In the present review, the term “nut” is used to identify the main culinary tree nuts (walnuts, hazelnuts, almonds, pistachios, cashews, Brazil nuts, pecans, and pine nuts) and peanuts (the seed of a legume plant that has a similar profile of nutrients to tree nuts) [11], although some of them do not achieve this botanical definition. Tree nuts and peanuts are nutrient-dense foods that constitute a rich source of unsaturated fatty acids, plant-based proteins, and dietary fiber, in addition to other components such as plant sterols, antioxidants, minerals, and vitamins [12]. Increasing evidence from epidemiological studies has supported the potential of daily nut consumption as a strategy for the primary prevention of obesity [13,14,15]. In addition to its potential effect in reducing the risk of obesity, emerging evidence has indicated that nut intake could contribute to the treatment of obesity. For instance, greater weight loss has been reported among overweight and obese individuals who consumed nut-enriched diets compared to their counterparts who consumed an isocaloric control diet [16]. Although the potential role of nuts in the prevention and treatment of obesity is not yet clear, some possible mechanisms have been suggested to explain these benefits, such as the incomplete absorption of the energy provided by nuts, increased satiety, and the control of hunger due to a large amount of fiber and a prebiotic effect on the gut microbiota [17,18,19]. However, because of their high energy and fat content, whether nuts are obesogenic foods is a common concern for which the answer remains elusive [20,21].

Some systematic reviews and meta-analyses have explored the effects of nut consumption on adiposity parameters, and their results are controversial. While some meta-analyses of specific nuts showed no significant changes in body weight (BW), body mass index (BMI), or waist circumference (WC) [22,23,24], others included different types of nuts and reported significant reductions in adiposity parameters among frequent nut consumers [25]. Furthermore, these reviews did not examine whether the association with adiposity parameters varied by the type of nut. Although nuts share many ingredients [12], the nutrient profile of each nut and their availability and form vary among them [26]. Thus, it is possible that not all nuts are associated in the same way with indicators of adiposity.

Therefore, our main research question is: “Does the scientific evidence based on RCTs support the claim that nut consumption improves weight and adiposity control in healthy individuals and reduces weight in people with obesity or chronic comorbidities?” As network meta-analysis (NMA) is the most suitable method to comparatively estimate the effects of different interventions (i.e., different types of nuts) on a health outcome (i.e., adiposity parameters) [27], we also comparatively assessed and integrated the effects of tree nuts and peanuts on BW, BMI, WC, and body fat percentage (BF%).

## 2. Methods

The present NMA was conducted based on the Cochrane Collaboration Handbook [28] and reported according to the Preferred Reporting Items for Systematic Reviews and Meta-Analyses, including network meta-analyses (PRISMA-NMA, checklist available in the Appendix A) [29]. We have registered the protocol of this NMA through PROSPERO (CRD42021232488).

### 2.1. Search Strategy

Two independent researchers (R.F.-R. and A.E.M.) systematically searched MEDLINE (via PubMed), EMBASE (via Scopus), and Web of Science (WOS) from database inception until 31 December 2020. The search strategy aimed to identify randomized controlled trials (RCTs) exploring the effects of nut consumption (i.e., almonds, cashews, hazelnuts, peanuts, pistachios, walnuts, and mixed nuts) on adiposity-related measures (BW, BMI, WC, and BF%). Our search strategy for the PubMed database combined the following terms: (nut OR almond OR hazelnut OR pistachio OR walnut OR cashew, OR nuts OR almonds OR hazelnuts OR pistachios OR walnuts OR cashews) AND (anthropometric OR weight OR “body weight” OR “fat” OR “fat mass” OR “fat percentage” OR BMI OR “body mass index” OR “waist circumference” OR “obese” OR “obesity” OR “overweight” OR “overweight” OR “body fat”) AND (RCT OR “randomized controlled trial” OR “controlled trial” OR “randomized controlled trial” OR “trial” OR “clinical trial” OR intervention). The complete search strategy for all databases is available in the (Appendix A). Moreover, we reviewed the list of references from previous systematic reviews and meta-analyses in the field. The EndNote X9 reference management software (Clarivate, The EndNote Team, PA, USA) was used to unify the searches of the different databases, exclude duplicate records, and select studies according to the inclusion and exclusion criteria.

### 2.2. Eligibility Criteria

In accordance with PICOS strategy, the eligible articles had to meet the following inclusion criteria: (1) type of studies: randomized, controlled trials with parallel or crossover design; (2) participants: adults ≥ 18 years old; (3) type of intervention: nut consumption (i.e., almonds, cashews, hazelnuts, peanuts, pistachios, walnuts, and mixed nuts, among others); in terms of dose and frequency, we also included trials that compared two or more different interventions in separate arm trials; (4) comparison: control groups with habitual diet, placebo or isocaloric snacks, low-fat diets or standard diets according to dietary guidelines, free of nut consumption; and (5) outcomes: adiposity-related measures (BW, BMI, WC, and BF%). When more than one study provided data referring to the same sample, the study providing more detailed data or with the largest sample size was selected. We excluded all other study designs, such as reviews, observational studies, abstracts, letters, and posters. No language restrictions were applied. The reasons for exclusion after full-text reading are available in the (Appendix A). When any discrepancy between the two independent researchers occurred, a third coauthor was consulted to resolve it (V.M.-V.).

### 2.3. Categorization of Available Evidence

Nut consumption interventions were determined for the following nut categories: (1) almonds, (2) Brazil nuts, (3) cashews, (4) hazelnuts, (5) macadamias, (6) peanuts, (7) pecans, (8) pistachios, (9) walnuts, and (10) mixed nuts. The categorization of the interventions was carried out based on the main nut-enriched diets included in the primary studies related to body composition parameters.

### 2.4. Data Extraction and Quality Assessment

All the data extraction and quality assessment processes (risk of bias and quality of evidence) were conducted by two independent researchers (R.F.-R. and A.E.M.), and when any discrepancy between them occurred, a third coauthor was consulted to resolve it (V.M.-V.). Microsoft Excel spreadsheets, designed specifically for this data extraction, were used. The following information for the included RCTs was extracted: name of the first author, publication year, country, study design (parallel or crossover), characteristics of the population (mean age, female percentage, mean baseline BW, BMI, WC, and BF%), health status of participants, total sample size and sample size by group for each trial arm, characteristics of the intervention (type and dose in terms of g/day, kcal/day, or energy percentage associated with nut consumption according to total calorie intake), comparison (type of diet), length of the intervention, outcome measures, and the main results. We considered outcome measures at baseline and at the end of each intervention period to conduct the analyses. The outcome measures related to adiposity were obtained in primary studies according to standardized and validated methods or with internally calibrated body composition analyzers. WC was assessed mostly with a measuring tape, as described in the literature, and BF% was assessed through bioelectrical impedance analysis or dual-energy X-ray absorptiometry.

The risk of bias was evaluated through the Cochrane Collaboration’s tool for assessing the risk of bias for RCTs (RoB2) [30]. Accordingly, five domains were evaluated: (1) randomization process bias; (2) deviations from the intended interventions bias; (3) missing outcome data bias; (4) outcome measurement bias; and (5) selection of the reported result bias. Finally, the overall risk of bias could be scored as “low risk of bias” when a low risk of bias was reached for all domains, “some concerns” when at least one domain was assessed as “some concerns” and any domain was assessed as “high risk of bias”, and “high risk of bias” when a high risk of bias was reached for at least one domain or the study had some concerns for multiple domains [30] To implement RoB2, the online tool https://www.riskofbias.info/welcome/rob-2-0-tool/current-version-of-rob-2 was used (accessed on 10 April 2021).

### 2.5. Grading the Quality of Evidence

The quality of the evidence contributing to the present network meta-analysis was evaluated for each outcome using the Grading of Recommendations Assessment, Development and Evaluation (GRADE) framework [31]. Each outcome (BW, BMI, WC, and BF%) could be scored as high, moderate, low, or very low evidence value, depending on the design of the studies, risk of bias, inconsistency, indirect evidence (related to indirect population, intervention, control, or outcomes), imprecision, and publication bias. Some factors may increase or decrease the quality score of the evidence. For instance, the risk of bias was −1 when <75% of the analyzed studies were at low risk of bias, heterogeneity was −1 when the heterogeneity shown by I^2^ value was >50%, and publication bias was −1 when it was reported. Two independent researchers (R.F.-R. and A.E.M.) assessed the quality of evidence, and disagreements were solved by consensus or with a third coauthor (V.M.-V.).

### 2.6. Dealing with Missing Data

According to the Cochrane Handbook [28], the SD values were calculated from SEs, 95% CIs, *p*-values, or *t*-statistics when these data were not directly available in the primary studies included.

### 2.7. Data Synthesis

We summarized the included studies qualitatively in a league table containing both direct and indirect comparisons. According to the PRISMA-NMA statement, we conducted NMA in the following steps. First, we showed the strength of the available evidence using a network diagram in which the size of the node indicates the number of trials included for each intervention, and the thickness of the continuous line connecting nodes is proportional to the number of trials directly comparing the two interventions [32].

After that, consistency was assessed by checking whether the intervention effects estimated from direct comparisons were consistent with those estimated by indirect comparisons. The Wald test was conducted, and due to the low statistical power, the side-splitting assessment was also used [33].

Once the consistency was evaluated, we conducted standard random-effects meta-analyses to determine the standardized mean differences (SMDs) of the interventions on the four adiposity outcomes using the Dersimonian and Laird method [34]. The I^2^ statistic was used to determine statistical heterogeneity, and, depending on its value, heterogeneity was considered to be not important (0% to 40%), moderate (30% to 60%), substantial (50% to 90%), or considerable (75% to 100%). We also took into account the corresponding *p*-value. Additionally, to determine the size and clinical relevance of heterogeneity, the τ^2^ statistic was calculated [35]. These results are displayed in forest plots and a league table. In addition, to determine the confidence and quality of the evidence, CINeMA software was used (available in the Appendix A) [36]. We assessed the consistency of direct and indirect effects in relation to clinically important SMDs with the CINeMA tool, and relative effect estimates below −0.20 and above 0.20 were considered clinically important for all outcomes [37].

The assumption of transitivity was evaluated to determine the potential effect modifiers among the available direct comparisons [33]; thus, we tested mean baseline differences of the control groups on the following potential confounders or effect modifiers: age, BW, BMI, WC, and BF%. In addition, for assessing the consistency of NMA estimates, direct and indirect estimates were compared using CINeMA software [38].

To control for potential confounders, we conducted subgroup analyses based on study design (parallel or crossover), type of nut (almond, hazelnut, walnut, and so on), health status of the participants (healthy, cardiovascular disease or at risk, dyslipidemia, metabolic syndrome, overweight or obesity, prediabetes or T2D) and type of control diet (hypocaloric diet, placebo diet, isocaloric diet, guideline-based diet, habitual diet, low-fat diet, or other control diets). Moreover, we conducted meta-regressions by length of the intervention, amount of nuts consumed, percentage of females, and baseline BMI. We also repeated subgroup analyses by type of nut in RCTs, designed to analyze whether nuts affect body weight reduction. To determine the potential clinical effects of nut consumption in people with overweight or obesity and in healthy participants, we conducted analyses based on the percentage of change, considering 5% of change in each baseline outcome a clinically significant difference and also taking into account the length of the intervention (<12 weeks vs. ≥12 weeks). Finally, we performed sensitivity analyses to assess the robustness of the summary estimates, and we assessed the publication bias and the small-study effect by a network funnel plot to determine the principle of symmetry [39]. Analyses were conducted in Stata 15.0 (Stata, College Station, TX, USA).

## 3. Results

### 3.1. Results of the Literature Search

The search identified a total of 5546 articles, and, after removing duplicates, 4002 were reviewed based on the title and abstract; 3760 were excluded. The full texts of the remaining 245 articles were reviewed, and 140 of them were excluded (reasons for exclusion are given in the Appendix A). Finally, 105 studies [7,8,9,16,40,41,42,43,44,45,46,47,48,49,50,51,52,53,54,55,56,57,58,59,60,61,62,63,64,65,66,67,68,69,70,71,72,73,74,75,76,77,78,79,80,81,82,83,84,85,86,87,88,89,90,91,92,93,94,95,96,97,98,99,100,101,102,103,104,105,106,107,108,109,110,111,112,113,114,115,116,117,118,119,120,121,122,123,124,125,126,127,128,129,130,131,132,133,134,135,136,137,138] were included in the present network meta-analysis (Figure 1).

### 3.2. Study Characteristics

Among the 105 included studies, 67 were conducted in parallel, and 38 were conducted in crossover designs. There were 103 comparisons available in 92 studies for BW (n = 6768), 72 comparisons in 69 studies for BMI (n = 2918), 60 comparisons in 53 studies for WC (n = 5045), and 33 comparisons in 30 studies for BF% (n = 1226). The population across the studies included healthy individuals as well as those with diagnoses of CVD (or at risk), dyslipidemia, metabolic syndrome, overweight or obesity, or prediabetes and type 2 diabetes. Participants from 16 different countries were followed for 2 to 240 weeks (mean: 18 weeks), and they had a mean age ranging from 18 to 74 years. Regarding the control groups, most studies used dietary guidelines according to the health status of participants, hypocaloric and low-fat diets, habitual or usual diets, and isocaloric snacks. Regarding the pairwise comparisons among nut-enriched diets, there were the following: for BW, three comparing walnut versus almond, and one comparing walnut versus cashew; for BMI, one comparing walnut versus almond, walnut versus cashew, and almond versus peanut; and for WC, one comparing walnut versus cashew. No direct comparisons of nuts were found for the BF% outcome. Further details are available in Appendix A.

### 3.3. Assessment of Transitivity and Consistency

The populations included in the control groups of the different nut interventions were similar in the baseline distribution of the potential effect modifiers and showed no significant differences in age (*p* = 0.12), BW (*p* = 0.38), BMI (*p* = 0.22), WC (*p* = 0.87), or BF% (*p* = 0.34) (Appendix A). Moreover, the consistency assumption of direct and indirect estimates was assessed using CINeMA software, showing no significant SMDs between them in each outcome with available mixed evidence (BW: χ^2^ statistic: 0.214 (2 degrees of freedom), *p*-value: 0.898 (Appendix A); BMI: χ^2^ statistic: 1.075 (3 degrees of freedom), *p*-value: 0.783 (Appendix A); and WC: χ^2^ statistic: 0.031 (1 degree of freedom), *p*-value: 0.860 (Appendix A) and BF%: statistics were not available because there was no comparison with both direct and indirect SMDs for BF% (Appendix A). Additionally, although the assumption of transitivity was achieved, this criterion should be interpreted with caution because we compared only some baseline characteristics of the studies and direct comparisons were scarcely available.

### 3.4. Risk of Bias

After the assessment by the RoB2 tool, the overall risk of bias was rated as “low risk of bias” in 28 studies, “some concerns” in 51 studies, and “high risk of bias” in 26 studies (Appendix A and Appendix A).

### 3.5. Network Analyses

Network diagrams show the relative amount of available evidence of direct comparisons among the different types of nuts consumed and control diets for each outcome (Figure 2A,B). Additionally, the network diagrams developed with the CINeMA web application depict the average risk of bias of comparisons within each outcome (Appendix A).

### 3.6. Outcomes

Body weight

Table 1 shows the comparisons in which nut-enriched diets did not modify BW compared to control diets.

Body mass index

Similarly, nut-enriched diets did not modify BMI compared to control diets (Table 2).

Waist circumference

Compared to control diets, almond-enriched diets were associated with a reduced WC (SMD: −0.15; 95% CI −0.27 to −0.02). Moreover, almond-enriched diets achieved significant reductions in WC compared with pistachio- (SMD: 0.30; 95% CI 0.04 to 0.55), mixed nuts- (SMD: 0.28; 95% CI 0.01 to 0.54), and hazelnut-enriched diets (SMD: 0.30; 95% CI 0.02 to 0.59) (Table 3).

Body fat percentage

When the nut-enriched diets were compared with the control diets, none were significantly associated with increased BF% (Table 4).

In addition, figures showing the network and pairwise meta-analysis SMDs, displayed as a league table, are available in the (Appendix A).

### 3.7. Clinical Percentage of Change

When considering the percentage of change in each outcome, there were significant changes for people with overweight and obesity on: BW, for nut (% change: −3.98; 95% CI −6.25 to −1.72) and control groups (% change: −3.37; 95% CI −5.79 to −0.96); BMI, for nut groups (% change: −3.29; 95% CI −6.21 to −0.38); and WC, for nut (% change: −4.10; 95% CI −7.27 to −0.94) and control groups (% change: −2.11; 95% CI −4.06 to −0.16). When the length of the nut intake intervention (<12 weeks vs. ≥12 weeks) was considered, the differences remained significant for the population with overweight and obesity and length ≥ 12 weeks on BW (% change: −4.94; 95% CI –7.30 to −2.58), BMI (% change: −4.34; 95% CI −8.13 to −0.55), and WC (% change: −4.87; 95% CI −8.98 to –0.77). Further details are available in Figure 3A,B and Appendix A.

### 3.8. Quality of Evidence

The quality of evidence for BW, BMI, WC, and BF% was assessed using the GRADE tool and rated as low or very low (Appendix A). Moreover, the CINeMA final report for each outcome, considering within-study bias, reporting bias, indirectness, imprecision, heterogeneity, incoherence, and confidence for each comparison, is described in the Appendix A.

### 3.9. Subgroup Analyses, Meta-Regressions, Sensitivity Analyses, and Small-Study Effects

Subgroup analyses based on study design showed a significant effect in favor of nut consumption for WC in parallel RCTs (SMD: −0.08; 95% CI −0.15 to −0.01). According to the type of nut, almond consumption was associated with a decrease in WC (SMD: −0.15; 95% CI −0.29 to −0.02), and hazelnut consumption was associated with an increase in WC (SMD: 0.30; 95% CI 0.02 to 0.59). In the subgroup analyses based on the health status of the participants, people with CVD (or at risk) and with overweight/obese conditions showed significant reductions in WC in the nut consumption groups compared to the control group (SMD: −0.06, 95% CI −0.11 to −0.01 and SMD: −0.17, 95% CI −0.32 to −0.02, respectively). Further details are available in the (Appendix A). In the subgroup analyses with only RCTs designed to assess whether nut consumption affected weight loss, almonds were associated with reduced BMI (SMD: −0.71; 95% CI −1.33, −0.08), hazelnuts with increased WC (SMD: 0.30; 95% CI 0.02, 0.59), and walnuts with reduced %BF (SMD: −0.85; 95% CI −1.20, −0.49). Overall, the SMD remained similar in subgroup analyses by the type of control diet (Appendix A). Meta-regressions by length, dosage of nuts, percentage of females, and baseline BMI did not significantly influence the SMD estimates for all outcomes (Appendix A).

Moreover, in the pairwise meta-analysis, when individual study data were removed from the analysis one at a time for each of the four outcomes, no changes were observed in the SMD estimates (Appendix A). Detailed results for small-study effects, according to Egger’s test and visual inspection of funnel plots, as well as the inconsistency assessment by I^2^ and τ^2^ statistics, are displayed in the (Appendix A).

## 4. Discussion

Although the most reported barrier to regular nut consumption is public concern related to weight gain, our data suggest that this concern is not sustained by scientific evidence. Data from our NMA, the first study that comparatively integrates the available evidence on the effects of tree nuts and peanuts on adiposity-related outcomes in adults, indicates that no type of nut increased BW, BMI, WC, or BF% (except for hazelnut with WC). In contrast, our analyses show a small significant decrease in WC with almond-enriched diets. When considering the health status of participants, significant reductions in WC were found for populations with or at risk of CVD as well as those with overweight or obesity. Studies conducted in people with overweight and obesity showed a significant percentage of change in both nut and control groups for BW, WC, and BMI (only nut groups); moreover, such differences were only significant with longer follow-up periods (>12 weeks). Finally, there was no influence of length, dosage of nuts, or female percentage on any adiposity outcome.

Our findings are consistent with a previous traditional meta-analysis that reported no significant effects of nut-enriched diets on BW, BMI, or WC [11,139]. In recent years, some studies examining isolated types of nuts have shown different findings. For instance, almond intake has been significantly associated with decreased BW and fat mass but not WC [22], while walnut [23] or cashew [24] consumption did not significantly modify adiposity measures. Furthermore, a recent meta-analysis of longitudinal studies and RCTs found significant reductions in BW, BMI, and WC when nut-enriched diets were compared with control diets [25]. A plausible explanation for these contradictory results might rely on the potentially different effects of nuts on adiposity parameters that could be partially solved with an NMA approach. Moreover, we must not forget that the “clustering effect” of a healthier lifestyle among nut consumers may also have some impact on adiposity [140].

We found that almonds had a significant effect on the reduction in WC. Accordingly, improvements in anthropometric measures after almond-enriched diets have been noticed in previous meta-analyses [22,25]. The small benefit shown by almonds might be explained through their nutrient profile [141,142]. They have the lowest lipid content of all nuts (55%), and the cell wall encapsulation of whole raw almonds during the digestion process may limit energy bioaccessibility [142]. In addition, fiber and polyphenols (i.e., tannins, proanthocyanins, and flavonoids) are bioactive compounds with antioxidant, antimicrobial, and antiviral properties that are particularly present in almond skin [141]. Furthermore, they favor gut microbiota and, consequently, energy metabolism [143].

Regarding the clinical percentage of change, none of the interventions (nut consumption or control) showed an increase or reduction in adiposity parameters exceeding the 5% considered meaningful in people with overweight or obesity and healthy participants. However, it is worth noting that only those studies conducted in people with overweight and obesity showed significant percentage reductions in both nut and control groups for BW, WC, and BMI (only nut groups), and that such differences were only significant with longer follow-up periods (>12 weeks). These results agreed with previous evidence in PREDIMED trials in a wide range of subjects that showed that nut consumption had an inverse association with excess weight and metabolic syndrome [144]. Although we have to interpret this with caution, the clinical message seems to be that nut consumption does not lead to a weight or adiposity increase in healthy/normal weight people or people with overweight/obesity. Moreover, nut intake as part of an intervention for people with excess weight could lead more easily to clinically meaningful reductions in BMI, and such reductions are maintained with longer follow-up interventions.

According to our subgroup analyses, nut-enriched diets show greater reductions in WC in populations with or at risk of CVD or with overweight or obesity than in healthy people. Because these participants are commonly under a multicomponent approach to reduce weight (i.e., restricted-energy intake diet, increased physical activity), it is reasonable to suppose that the effects on adiposity could be due to synergistic interaction. Although the average length of the intervention was longer in subgroups with CVD (or at risk) and with overweight or obesity compared to other intervention groups (average weeks: 26 vs. 18, respectively), these subgroups represented more than half (65.4%) of the total participants in the WC outcome. Because of the importance of this significant WC reduction from a clinical perspective, studies should be undertaken to determine the independent effects of nut consumption in people at risk of cardiometabolic disorders.

Some plausible mechanisms underlying our results should be mentioned. Although nuts are energy-dense foods, their metabolizable energy is lower than expected due to the indigestible and encapsulated wall components of their skin [18]. Accordingly, oral processing effort, incomplete mastication, and unsaturated fatty acid content enhance resting energy expenditure and thermogenesis, while dietary fiber slows carbohydrate absorption [19]. Moreover, their polyphenol content (mainly in unprocessed skins) impacts gut microbiota health, boosting the growth of beneficial bacteria that may prevent weight gain [17]. Considering the above and that roasting processes cause mastication to result in smaller particles that could increase the energy extracted from nuts, it seems that the best way to consume nuts without concerns about weight gain might be in their whole raw unprocessed form [18].

The following limitations of the present NMA should be considered. First, the quality of evidence of our findings was graded as low to very low. Second, the small number of available trials for direct comparisons as well as for the different types of nuts may have influenced our findings. Third, although we considered the length of intervention and this feature did not significantly influence our findings, there are some studies with relatively short-term (mean 18 weeks) interventions. This could have impeded the ability to show adiposity changes that could potentially occur in the long term. Fourth, some included studies were conducted to maintain a stable BW during the intervention, which could have potentially affected our estimates. Considering this, high-quality and well-designed trials are needed to determine whether the potential benefits of some nuts, such as almonds, may produce more than a small clinical impact on adiposity-related outcomes in long-term interventions. Last, although our data seem to be robust because they have been extracted from RCTs, we cannot state to what extent the effects of nut intake on adiposity can be explained by other energy-balance-related behaviors, such as physical activity, energy eating patterns, overall diet quality, mode of nut preparation (i.e., raw, roasted) or timing of food intake.

Some methodological aspects differentiate the present NMA from previous reviews and need to be acknowledged. First, the effects of different nuts on body weight had not been comparatively analyzed. While other reviews explored specific nuts [22,23,24], we included studies on all culinary nuts and compared their effects using NMA methodology. This has allowed us to, for example, draw specific conclusions about the potential benefits of almond consumption that were not observed for other nuts. Second, we have analyzed as an outcome the four main indicators of body adiposity at the same time, which allows us to gather and comprehensively synthesize the evidence on the relationship between nut consumption and adiposity. Thus, our take-home message allows us to affirm not only that nuts do not increase weight but also that they do not negatively affect other adiposity-related cardiometabolic markers such as waist circumference and body fat percentage. Lastly, our analyses thoroughly explored the effects of many potential confounders of the relationship between nuts and adiposity. Considering the subgroup analyses, we were able to provide information disaggregated by study design, type of nut, health and nutritional status of participants, and type of control used for comparison with nut intake. In addition, we estimated SMD in meta-regression models, adjusted for time of trial duration, daily nut dose, % of women among participants, and baseline BMI.

## 5. Conclusions

Tree nuts and peanuts are rich sources of unsaturated fatty acids, plant proteins, vitamins, and minerals [12] and are globally considered high-energy foods with appreciated sensorial, nutritional, and health attributes [145]. Based on our findings, health professionals might recommend the consumption of nuts with the certainty that this type of food does not have a negative influence on adiposity parameters and, therefore, does not increase the risk of obesity.

In summary, to our knowledge, none of the previous reviews have examined the relationship between nuts and adiposity so comprehensively. In addition to confirming the results of previous reviews on the absence of weight gain when eating nuts, ours is innovative in pointing out that some nuts can even help you lose weight and reduce waist circumference. These results have scientific, clinical, and public health implications. First, they indicate the need for future RCTs to analyze the contributing role of nut consumption as part of strategies for reducing body weight and controlling adiposity. Second, health professionals should not only not be worried about the risk of their patients gaining weight when consuming nuts, but they can even recommend the consumption of these foods to help control body weight. Finally, combining the well-known beneficial effects of nuts for cardiovascular health with a possible additional effect in the control of adiposity, the importance of promoting the consumption of nuts should be reinforced through their inclusion in the guidelines for healthy eating for the general population.

## Figures and Tables

**Figure 1 nutrients-13-02251-f001:**
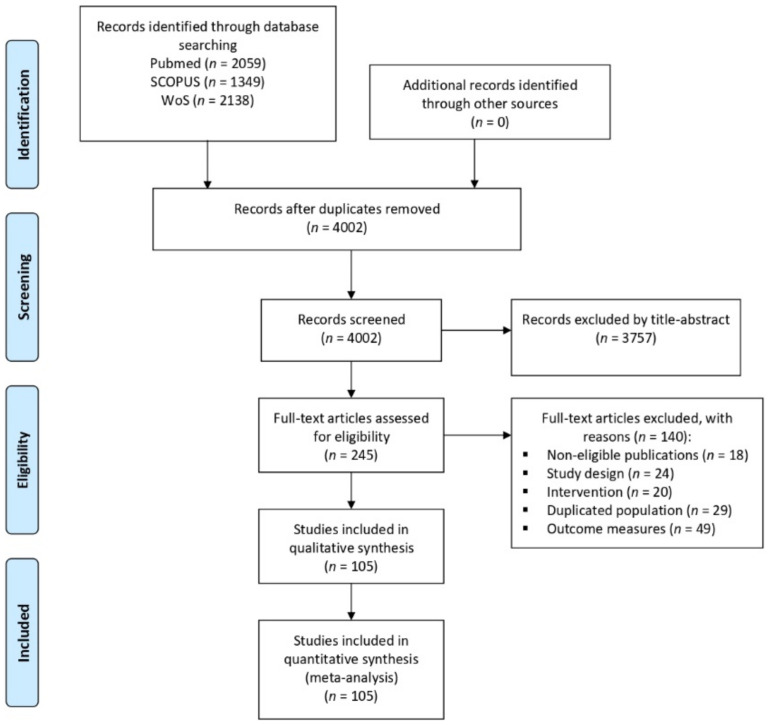
Flow diagram of the study selection process.

**Figure 2 nutrients-13-02251-f002:**
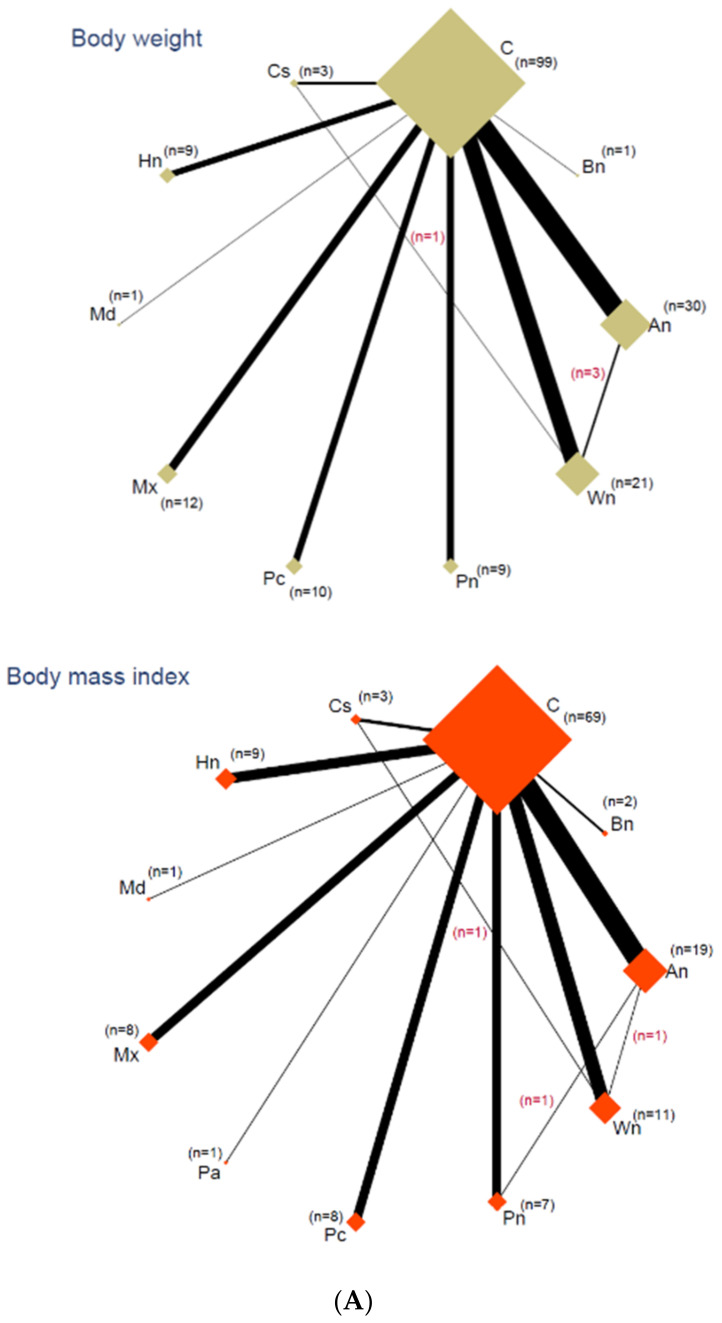
(**A**) Network diagram for body weight and BMI. The size of the diamond is proportional to the number of trials included of each intervention, and the line width corresponds to trials directly comparing the two interventions. (**B**). Network diagram for WC and BF%. The size of the diamond is proportional to the number of trials included of each intervention, and the line width corresponds to trials directly comparing the two interventions. **Abbreviations**: C: control; Bn: Brazil nut; An: almond; Wn: walnut; Pn: peanut; Pc: pistachio; Mx: mixed nut; Md: macadamia; Hn: hazelnut; Cs: cashew; BMI: body mass index; WC: waist circumference; BF%: body fat percentage.

**Figure 3 nutrients-13-02251-f003:**
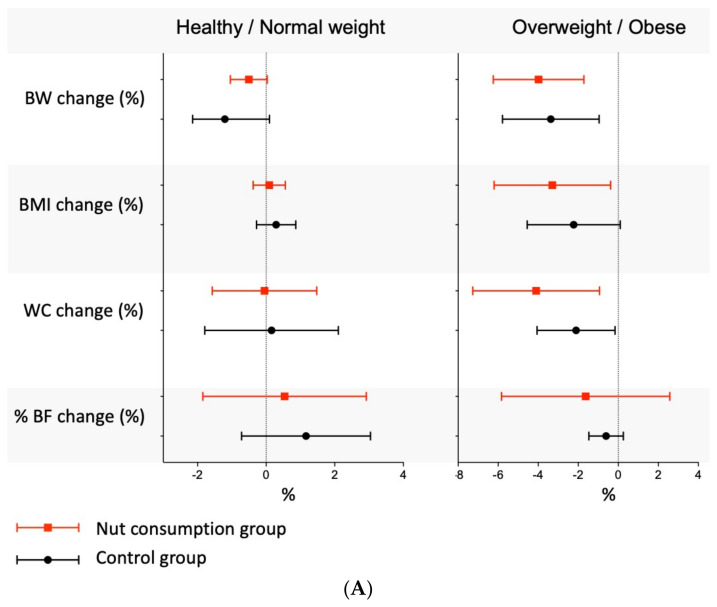
(**A**) Percentage of change in each outcome for healthy/normal weight and people with overweight/obesity. (**B**). Percentage of change in each outcome for healthy/normal weight and people with overweight/obesity considering the length of time among the nut interventions. **Abbreviations**: BW: body weight; BMI: body mass index; WC: waist circumference; BF%: body fat percentage.

**Table 1 nutrients-13-02251-t001:** Standardized mean differences (SMDs) and 95% CI on body weight (n = 103 observations from 99 RCTs). The upper and colored right triangle gives the SMDs from pairwise comparisons (column intervention relative to row), lower left triangle refers to the SMDs from the network meta-analysis (row intervention relative to column).

	Control	Walnuts	Almonds	Mixed	Pistachios	Peanuts	Hazelnuts	Cashews	Brazil Nuts	Macadamia
Control		0.03 (−0.05 to 0.11)	−0.07 (−0.16 to 0.02)	0.02 (−0.03 to 0.07)	<0.01 (−0.15 to 0.16)	0.15 (−0.03 to 0.33)	0.07 (−0.12 to 0.25)	0.03 (−0.18 to 0.24)	−0.07 (−1.01 to 0.58)	−0.12 (−0.47 to 0.23)
Walnuts	0.01 (−0.08 to 0.09)		<−0.01 (−0.42 to 0.41)	NA	NA	NA	NA	0.03 (−0.58 to 0.64)	NA	NA
Almonds	−0.06 (−0.15 to 0.02)	−0.07 (−0.19 to 0.05)		NA	NA	NA	NA	NA	NA	NA
Mixed	0.02 (−0.02 to 0.07)	0.02 (−0.08 to 0.11)	0.09 (−0.01 to 0.19)		NA	NA	NA	NA	NA	NA
Pistachios	<0.01 (−0.16 to 0.17)	<−0.01 (−0.18 to 0.19)	0.07 (−0.11 to 0.26)	−0.01 (−0.19 to 0.16)		NA	NA	NA	NA	NA
Peanuts	0.05 (−0.11 to 0.20)	0.04 (−0.14 to 0.21)	0.11 (−0.07 to 0.29)	0.02 (−0.14 to 0.19)	0.04 (−0.19 to 0.26)		NA	NA	NA	NA
Hazelnuts	0.10 (−0.08 to 0.28)	0.09 (−0.11 to 0.29)	0.16 (−0.04 to 0.36)	0.07 (−0.11 to 0.26)	0.09 (−0.15 to 0.33)	0.05 (−0.18 to 0.29)		NA	NA	NA
Cashews	0.04 (−0.16 to 0.24)	0.03 (−0.18 to 0.24)	0.10 (−0.11 to 0.32)	0.02 (−0.19 to 0.22)	0.03 (−0.19 to 0.26)	−0.01 (−0.26 to 0.25)	−0.06 (−0.32 to 0.21)		NA	NA
Brazil nuts	−0.05 (−1 to 0.90)	−0.06 (−1.01 to 0.90)	0.01 (−0.95 to 0.96)	−0.07 (−1.03 to 0.88)	−0.06 (−1.02 to 0.91)	−0.10 (−1.06 to 0.87)	−0.15 (−1.12 to 0.82)	−0.09 (−1.06 to 0.88)		NA
Macadamia	−0.11 (−0.46 to 0.24)	−0.12 (−0.48 to 0.24)	−0.05 (−0.40 to 0.31)	−0.13 (−0.48 to 0.22)	−0.12 (−0.50 to 0.26)	−0.16 (−0.54 to 0.22)	−0.21(−0.60 to 0.18)	−0.15 (−0.55 to 0.25)	−0.06 (−1.07 to 0.95)	

CI: confidence interval; NA: Not available; RCTs: randomized, controlled trials; SMD: standardized mean difference.

**Table 2 nutrients-13-02251-t002:** Standardized mean differences (SMDs) and 95% CI on BMI (n = 72 observations from 69 RCTs). The upper and colored right triangle gives the SMDs from pairwise comparisons (column intervention relative to row), lower left triangle refers to the SMDs from the network meta-analysis (row intervention relative to column).

	Control	Walnuts	Almonds	Mixed	Pistachios	Peanuts	Hazelnuts	Cashews	Brazil nuts	Macadamia	Pecan
Control		0.04 (−0.06 to 0.14)	−0.08 (−0.23 to 0.06)	−0.05 (−0.20 to 0.10)	<0.01 (−0.17 to 0.19)	0.06 (−0.11 to 0.22)	0.07 (−0.12 to 0.25)	<0.01 (−0.21 to 0.21)	−0.13 (−0.70 to 0.44)	−0.07 (−0.41 to 0.28)	0 (−0.9 to 0.9)
Walnuts	0.02 (−0.08 to 0.12)		0.27 (−0.44 to 0.98)	NA	NA	0.13 (−0.66 to 0.92)	NA	−0.04 (−0.57 to 0.65)	NA	NA	NA
Almonds	−0.04 (−0.15 to 0.07)	−0.06 (−0.20 to 0.08)		NA	NA	NA	NA	NA	NA	NA	NA
Mixed	0.05 (−0.10 to 0.20)	0.03 (−0.15 to 0.20)	0.09 (−0.10 to 0.27)		NA	NA	NA	NA	NA	NA	NA
Pistachios	0.03 (−0.15 to 0.21)	0.01 (−0.19 to 0.21)	0.07 (−0.14 to 0.28)	−0.02 (−0.25 to 0.22)		NA	NA	NA	NA	NA	NA
Peanuts	<−0.01 (−0.17 to 0.17)	−0.02 (−0.21 to 0.18)	0.04 (−0.16 to 0.24)	−0.04 (−0.27 to 0.18)	−0.03 (−0.28 to 0.22)		NA	NA	NA	NA	NA
Hazelnuts	0.09 (−0.10 to 0.28)	0.07 (−0.14 to 0.28)	0.13 (−0.09 to 0.35)	0.04 (−0.19 to 0.28)	0.06 (−0.20 to 0.32)	0.09 (−0.16 to 0.34)		NA	NA	NA	NA
Cashews	−0.10 (−0.30 to 0.10)	−0.12 (−0.34 to 0.10)	−0.06 (−0.29 to 0.17)	−0.14 (−0.39 to 0.10)	−0.13 (−0.40 to 0.14)	−0.10 (−0.36 to 0.16)	−0.19 (−0.46 to 0.08)		NA	NA	NA
Brazil nuts	−0.16 (−0.72 to 0.41)	−0.18 (−0.75 to 0.40)	−0.12 (−0.70 to 0.46)	−0.20 (−0.79 to 0.39)	−0.19 (−0.78 to 0.41)	−0.16 (−0.75 to 0.43)	−0.25 (−0.85 to 0.35)	−0.06 (−0.66 to 0.54)		NA	NA
Macadamia	−0.09 (−0.43 to 0.26)	−0.11 (−0.47 to 0.25)	−0.05 (−0.41 to 0.31)	−0.14 (−0.51 to 0.24)	−0.12 (−0.51 to 0.27)	−0.09 (−0.48 to 0.29)	−0.18 (−0.57 to 0.21)	<0.01 (−0.39 to 0.41)	0.07 (−0.60 to 0.73)		NA
Pecan	<0.01 (−0.9 to 0.9)	−0.02 (−0.92 to 0.88)	0.04 (−0.87 to 0.94)	−0.05 (−0.96 to 0.86)	−0.03 (−0.95 to 0.89)	<−0.01 (−0.92 to 0.91)	−0.09 (−1.01 to 0.83)	0.10 (−0.82 to 1.02)	0.16 (−0.91 to 1.22)	0.09 (−0.87 to 1.05)	

CI: confidence interval; NA: Not available; RCTs: randomized, controlled trials; SMD: standardized mean difference; WC: waist circumference.

**Table 3 nutrients-13-02251-t003:** Standardized mean differences (SMDs) and 95% CI on WC (n = 60 observations from 59 RCTs). The upper and colored right triangle gives the SMDs from pairwise comparisons (column intervention relative to row), lower left triangle refers to the SMDs from the network meta-analysis (row intervention relative to column).

	Control	Walnuts	Almonds	Mixed	Pistachios	Peanuts	Hazelnuts	Cashews	Brazil nuts	Macadamia
Control		<−0.01 (−0.12 to 0.11)	−**0.15** (−0.29 to −0.02)	−0.15 (−0.35 to 0.05)	0.14 (−0.06 to 0.35)	−0.05 (−0.22 to 0.11)	**0.30** (0.02 to 0.59)	<−0.01 (−0.22 to 0.20)	0.07 (−0.49 to 0.64)	−0.07 (−0.42 to 0.28)
Walnuts	<0.01 (−0.11 to 0.12)		NA	NA	NA	NA	NA	0.06 (−0.55 to 0.67)	NA	NA
Almonds	−**0.15** (−0.27 to −0.02)	−0.15 (−0.32 to 0.01)		NA	NA	NA	NA	NA	NA	NA
Mixed	−0.12 (−0.27 to 0.02)	−0.02 (−0.62 to 0.59)	0.02 (−0.16 to 0.21)		NA	NA	NA	NA	NA	NA
Pistachios	0.16 (−0.07 to 0.38)	0.15 (−0.10 to 0.40)	**0.30** (0.04 to 0.55)	**0.28** (0.01 to 0.54)		NA	NA	NA	NA	NA
Peanuts	−0.01 (−0.20 to 0.19)	−0.01 (−0.24 to 0.21)	0.14 (−0.09 to 0.37)	0.11 (−0.13 to 0.36)	−0.16 (−0.46 to 0.14)		NA	NA	NA	NA
Hazelnuts	0.18 (−0.12 to 0.49)	0.18 (−0.15 to 0.51)	0.33 (−0.001 to 0.66)	0.30 (−0.03 to 0.64)	0.03 (−0.35 to 0.41)	0.19 (−0.17 to 0.55)		NA	NA	NA
Cashews	<0.01 (−0.24 to 0.24)	<−0.01 (−0.27 to 0.26)	0.15 (−0.12 to 0.41)	0.12 (−0.15 to 0.40)	−0.15 (−0.48 to 0.17)	<0.01 (−0.29 to 0.31)	−0.18 (−0.57 to 0.21)		NA	NA
Brazil nuts	−0.01 (−0.60 to 0.58)	−0.02 (−0.62 to 0.59)	0.14 (−0.46 to 0.74)	0.11 (−0.49 to 0.72)	−0.17 (−0.80 to 0.46)	<−0.01 (−0.62 to 0.61)	−0.19 (−0.86 to 0.48)	−0.01 (−0.65 to 0.63)		NA
Macadamia	−0.07 (−0.47 to 0.33)	−0.07 (−0.50 to 0.35)	0.08 (−0.33 to 0.48)	0.05 (−0.36 to 0.47)	−0.23 (−0.68 to 0.22)	−0.06 (−0.50 to 0.37)	−0.25 (−0.76 to 0.26)	−0.07 (−0.54 to 0.40)	−0.06 (−0.78 to 0.66)	

CI: confidence interval; NA: Not available; RCTs: randomized, controlled trials; SMD: standardized mean difference; SMD in bold: statistically significant; WC: waist circumference.

**Table 4 nutrients-13-02251-t004:** Standardized mean differences (SMDs) and 95% CI on BF% (n = 33 observations from 33 RCTs). The upper and colored right triangle gives the SMDs from pairwise comparisons (column intervention relative to row), and lower left triangle refers to the SMDs from the network meta-analysis (row intervention relative to column).

	Control	Walnuts	Almonds	Mixed	Pistachios	Peanuts	Hazelnuts	Brazil Nuts
Control		−0.16 (−0.40 to 0.09)	0.04 (−0.11 to 0.20)	−0.23 (−0.52 to 0.06)	−0.07 (−0.73 to 0.68)	−0.16 (−0.49 to 0.18)	0.05 (−0.16 to 0.25)	−0.02 (−0.73 to 0.69)
Walnuts	−0.19 (−0.41 to 0.03)		NA	NA	NA	NA	NA	NA
Almonds	0.03 (−0.18 to 0.24)	0.22 (−0.09 to 0.52)		NA	NA	NA	NA	NA
Mixed	−0.23 (−0.56 to 0.10)	−0.04 (−0.44 to 0.35)	−0.26 (−0.65 to 0.13)		NA	NA	NA	NA
Pistachios	−0.06 (−0.71 to 0.59)	0.13 (−0.56 to 0.82)	−0.09 (−0.77 to 0.60)	0.17 (−0.56 to 0.90)		NA	NA	NA
Peanuts	−0.19 (−0.59 to 0.22)	<0.01 (−0.45 to 0.46)	−0.21 (−0.67 to 0.24)	0.05 (−0.47 to 0.56)	−0.13 (−0.89 to 0.64)		NA	NA
Hazelnuts	0.03 (−0.24 to 0.29)	0.21 (−0.13 to 0.56)	<−0.01 (−0.34 to 0.34)	0.26 (−0.16 to 0.68)	0.09 (−0.62 to 0.79)	0.21 (−0.27 to 0.69)		NA
Brazil nuts	−0.03 (−0.85 to 0.79)	0.16 (−0.69 to 1.00)	−0.06 (−0.90 to 0.79)	0.20 (−0.68 to 1.08)	0.03 (−1.02 to 1.08)	0.16 (−0.76 to 1.07)	−0.06 (−0.92 to 0.80)	

BF%: Body fat percentage; CI: confidence interval; NA: Not available; RCTs: randomized, controlled trials; SMD: standardized mean difference.

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
