# Peer review of "The Relationship of Tree Nuts and Peanuts with Adiposity Parameters: A Systematic Review and Network Meta-Analysis"

_nutrients, 2021, doi:10.3390/nu13072251_

Round 1

Reviewer 1 Report

Fernández-Rodríguez and colleagues have conducted a systematic review with network meta-analysis on the relationship between consumption of culinary tree nuts and peanuts and obesity outcomes. Their analysis was done carefully and using best practice. However, I have some issues with the presentation of the data, and the interpretation.

Major points:

  • You included only randomised, controlled trials in your analysis. This was the correct approach, but from your abstract and your Eligibility criteria this was not clear. In line 91, please change “clinical trials with parallel or crossover design” to “randomized, controlled trials with parallel or crossover design.” This is important, because RCTs can assign causality, whereas cohort studies etc cannot.
  • It was not clear from the methods whether the effect sizes were MD or SMD. In your protocol on PROSPERO, you state that you will only report on SMDs, but Table 1 is labelled as “pooled mean differences”. Please update your methods section to state whether mean differences or standardised mean differences were reported (they appear to be SMDs but I need to be sure). Also the title of Table 1 is not complete.
  • The upper CIs are missing from all the effect sizes in Table 1. I need to see these.
  • My main issue with this analysis is the interpretation of the results. In your network MA, you combined studies that compared nut consumption with habitual diet, placebo (whatever that could be), isocaloric snacks, low-fat diets, standard diets and nut-free diets. Then you checked for markers of obesity. The problem is that you provided no plausible explanation for the use of nuts as a preventer of, or a treatment for, obesity. Is this supposed to be due to satiety? This matters, because if you think nuts may work by reducing hunger, you want to use isocaloric snacks or at least an isocaloric diet as a comparator.
  • You must provide an hypothesis for the role of nuts in treating obesity. Even your introduction states “Increasing evidence from epidemiological studies has supported the potential of daily nut consumption as a strategy for the primary prevention of obesity” but then you do not use studies aimed at prevention of obesity, but rather whether people LOST weight or not.
  • Or was it that your study was aiming to test whether nuts would INCREASE body weight etc? It is very unclear from your introduction exactly what your hypothesis is that you were testing. I find it useful to define a PICOTS – it forces you to ask a scientific question that you then answer. For example: In adults of normal weight, does consumption of any type of culinary nuts, compared with an isoenergetic diet without nuts, lower body weight, BMI etc over a period of at least 4 weeks in an outpatient setting? OR In overweight but not obese adults, does a diet high in nuts, compared with a diet low in nuts, prevent weight gain over a period of 6 months in an inpatient setting? What is your actual question?
  • You did not define what a clinically significant difference in body weight, BMI, waist circumference or % body fat is. For example, it is generally accepted that an increase in or reduction of 5% of baseline bodyweight is considered to be clinically significant. You need to interpret your results in relation to the relevance to the physician and the patient.
  • You also did not state whether each study had the aim of weight loss or not. If the primary aim of a study was, for example, change in LDL, then they may have tried to keep the calories the same between the groups, and you will see no change in body weight etc.
  • You must provide a subgroup meta-analysis for each nut type by control type, or at least performed meta-regression on this.
  • You must provide a subgroup meta-analysis for each nut type, where only weight-loss studies are included.
  • You must provide a meta-regression or subgroup analysis with baseline BMI as a covariate.
  • Your current study shows that for body weight: none of the comparisons was statistically significant (either from the NMA or pairwise) – at least this is what I suspect (I can’t see the upper CI for any comparison). For BMI: same as body weight. Waist circumference: Almonds were better than control and pistachios, Mixed nuts were better than pistachios. Percent body fat: no differences.
  • This is my biggest problem: you conducted SUCRA plots for outcomes where none of the interventions was statistically significantly different from the control. This makes no sense. By definition, none of these interventions (or control) is different from any other! By “ranking” things that don’t differ from one another, you are giving a false interpretation of your results.
  • My suggestion would be to rethink the point of your analysis. You have so many included studies, but I think this has, surprisingly, just made your analysis lack meaning. You need to either focus your question much more narrowly, or else do much more detailed subgroup analysis/meta-regression using all the covariates as your disposal. Your supplementary material was 90 pages long, and I expected to see all these analyses in there, but was disappointed.
  • Please don’t be too disappointed. You have done an enormous amount of work and it is not wasted. Your search terms were good, your methodology is good, you have plenty of studies. Use this to go back and ask your questions first, then design your analyses around your questions.

Minor points:

  • It may seem pedantic, but there is a difference between the culinary and botanical definition of nuts. I think it is important for people to understand that not many culinary “nuts” are not botanically nuts (such as almonds, pecan, pistachios, walnuts, Brazil nuts, pine nuts, peanuts and coconuts). It is not important for your analysis, but it is an opportunity to educate your readers. Please update your introduction to at least mention this.
  • The “Na”s in the tables should be NA or N/A.
  • You did not state what technology was used for independent inclusion/exclusion, data extraction, or quality assessment.

Author Response

Please see the attachment response.

Reviewer 2 Report

This paper conducts a network meta-analysis that seeks to compare the effect of different nut types body measures such as weight, BMI etc.  Most of the paper is clearly written, although sometimes much more information is needed.  I have some comments and suggestions below.

One major query I have is what the author's expected, and why this was important.  Other meta-analyses have been largely unsuccessful at showing an effect of nut intake on measures such as weight (or is they have then the effect has been small), so by combining many studies with reasonable sample sizes that lacked evidence of effectiveness against a control group even after pooling, was it anticipated that differences in but types would be found?  There are a lot of comparisons as well, and anything detected as significant looks small.  Are they clinically meaningful, by chance etc.?  Also, in at least one case the evidence found in the direct comparison, is not supported by the findings of the network meta-analysis combing both direct and indirect evidence.  The conclusion doesn't seem to add anything to what has been reported before.  E.g. "Based on our findings, health professionals might recommend the consumption of nuts with the certainty that this type of food does not have a negative influence on adiposity parameters and therefore does not increase 433 the risk of obesity."  I also this statement is flawed, how can this be made with "certainty"?  This type of analysis can be used to inform future research and not makes statements regarding causality.  Although it does appear that existing evidence in controlled trials suggests this.  In summary, I think there needs to be much work done to convince readers that this work adds value to the existing literature and make it very clear what it is the findings truly mean for future research.  I have some further comments below.

Is the assertion that the transitivity assumption was met too strong?  I think it is, with hardly any direct comparisons available, I think this is difficult to ascertain even if baseline measures do not differ much.  There is just too much that is unknown to make the claim solely on some baseline characteristics. This should be a noted limitation.  And, with some direct comparisons available, what were the results of testing the transitivity assumption comparing the direct and indirect?  It would be good to make this

Tables not formatted correctly meaning that not everything is displayed.  Perhaps a problem with conversion to pdf?

All tables labelled as Table 1.  Do they collectively form one table?  Also, in the tables it needs to be made clear that the lower triangle is for the comparisons resulting from the network meta-analysis and the upper for the direct comparisons.    In fact, in general I think there has to be much more guidance for the reader.  Network meta-analyses are no so common place that those interested in the results will understand what they are looking at.

Are the rankings a cause for concern?  The fact they can be quite different across the different outcomes when the outcomes are related is unusual.  Or are the differences in outcomes enough that the difference in ranking can be expected?  Although it is also isn't clear what the rankings are really for in many cases, with such small and insignificant effects found.

Please check the numbers in the supplementary tables.  E.g., means and SD of age, but one number only.  And values are incorrect, such as mean age of 324 for Abdrabalnabi. BF% of over 100 etc.  Also, the CIs in Table C6.

Author Response

Please see the attachment response.

Round 2

Reviewer 1 Report

You have addressed all my comments to my satisfaction. The new text needs some proofreading for English, but is otherwise fine to publish.

Reviewer 2 Report

I think that the authors have done a good job addressing my main concerns.